# High-Quality Single-Crystalline *β*-Ga_2_O_3_ Nanowires: Synthesis to Nonvolatile Memory Applications

**DOI:** 10.3390/nano11082013

**Published:** 2021-08-06

**Authors:** Chandrasekar Sivakumar, Gang-Han Tsai, Pei-Fang Chung, Babu Balraj, Yen-Fu Lin, Mon-Shu Ho

**Affiliations:** 1Department of Physics, National Chung Hsing University, Taichung 40227, Taiwan; chandruphysics1995@gmail.com (C.S.); pfchung123@gmail.com (P.-F.C.); yenfulin@nchu.edu.tw (Y.-F.L.); 2Innovation and Development Center of Sustainable Agriculture (IDCSA), National Chung Hsing University, Taichung 40227, Taiwan; 3Institute of Nanoscience, National Chung Hsing University, Taichung 40227, Taiwan; a0972000662@gmail.com; 4Department of Electrical Engineering, National Tsing Hua University, Hsinchu 300044, Taiwan; babubalrajr@gmail.com

**Keywords:** *β*-Ga_2_O_3_ nanowire, VLS growth, resistive memory, nonvolatile memory, SCLC

## Abstract

One of the promising nonvolatile memories of the next generation is resistive random-access memory (ReRAM). It has vast benefits in comparison to other emerging nonvolatile memories. Among different materials, dielectric films have been extensively studied by the scientific research community as a nonvolatile switching material over several decades and have reported many advantages and downsides. However, less attention has been given to low-dimensional materials for resistive memory compared to dielectric films. Particularly, *β*-Ga_2_O_3_ is one of the promising materials for high-power electronics and exhibits the resistive switching phenomenon. However, low-dimensional *β*-Ga_2_O_3_ nanowires have not been explored in resistive memory applications, which hinders further developments. In this article, we studied the resistance switching phenomenon using controlled electron flow in the 1D nanowires and proposed possible resistive switching and electron conduction mechanisms. High-density *β*-Ga_2_O_3_ 1D-nanowires on Si (100) substrates were produced via the VLS growth technique using Au nanoparticles as a catalyst. Structural characteristics were analyzed via SEM, TEM, and XRD. Besides, EDS, CL, and XPS binding feature analyses confirmed the composition of individual elements, the possible intermediate absorption sites in the bandgap, and the bonding characteristics, along with the presence of various oxygen species, which is crucial for the ReRAM performances. The forming-free bipolar resistance switching of a single *β*-Ga_2_O_3_ nanowire ReRAM device and performance are discussed in detail. The switching mechanism based on the formation and annihilation of conductive filaments through the oxygen vacancies is proposed, and the possible electron conduction mechanisms in HRS and LRS states are discussed.

## 1. Introduction

Gallium oxide is one of the ultra-wide bandgap semiconductors with a bandgap of 4.4–4.9 eV depending on its crystal structure, such as monoclinic (*β*-Ga_2_O_3_), corundum (*α*-Ga_2_O_3_), hexagonal (*ε*-Ga_2_O_3_), defective spinel (*γ*-Ga_2_O_3_), and bixbyite (*δ*-Ga_2_O_3_) [1,2,3,4]. However, the “*β*” phase Ga_2_O_3_ has had much attention as an excellent material for gas sensors [5], power devices [6], catalysis [7], and transparent conducting materials [8] due to its availability of single-crystalline substrates, high thermal and chemical stability, lower power loss, high electric breakdown field strength, controllability of conductivity, and high transparency in the visible region [9,10]. In addition, recently, Alhalaili et al. (2020) studied the morphological characteristics of *β*-Ga_2_O_3_ nanowires grown on different substrates with different temperatures and reported the suitable conditions to grow high-density unform *β*-Ga_2_O_3_ nanowires on Si substrates [11]. The *p*-type doping is still a significant obstacle to the device application. To extend the prospects of *β*-Ga_2_O_3_-based nano-optoelectronic devices, it is necessary to obtain both n-type and p-type *β*-Ga_2_O_3_ nanomaterials, which would facilitate the fabrication of PN junctions, as well as transistors, by simply crafting P and N wells in the semiconductor by an efficient ion implantation process, similar to the conventional method used in the silicon-based semiconductor fabrication processes. Commonly, undoped *β*-Ga_2_O_3_ exhibits n-type conductivity due to the donor-related natural impurities and defects [12,13,14,15].

Resistive random-access memory (ReRAM) with a simple sandwich structure of metal/insulator/metal (MIM) is one of the emerging nonvolatile memory technologies that has excellent features such as faster writing speed, smaller device feature size, lower programming voltage, and multistate memory compared to its competing memories such as dynamic random-access memory (DRAM), phase-change random-access memory (PCRAM), and conductive bridge random-access memory (CBRAM) [16]. A large number of research articles discussing the resistive memory of metal oxides have been published every year ever since the need for next-generation memory devices swelled. Meanwhile, understanding the fundamental physics behind resistive switching is also crucial, and from various density functional theoretical (DFT) studies and experimental studies, the switching mechanism of a metal oxide ReRAM device has been explained [17,18,19]. Mostly, in metal oxides, the oxygen vacancy filament mechanism dominates compared to other switching mechanisms, and oxygen filaments-dominated resistive memory is a prominent and highly reliable candidate for the next-generation memory [20,21]. Dielectric thin films as a resistive switching material have been studied extensively through the scientific community [16]. However, low-dimensional *β*-Ga_2_O_3_ nanostructures, including nanowires, nanorods, nanobelts, nanoribbons, and nanoflakes, have massive potential in various semiconductor applications that have not been explored fully [22,23]. Among these structures, nanowires are considered excellent materials due to their controllability in growth, electron flow, and ease of obtaining a high-quality, as well as single-crystalline, structure, making them ideal candidates for device fabrication. The use of such materials is determined by a number of elements from a theoretical perspective. The degree of freedom, the defect arrangement, and the electrical state are the most critical factors. Having the least degrees of freedom provides for greater control over defect arrangements, limiting the leakage current to a minimum during device operation via an external electric field. Furthermore, when the material is scaled down to nanometers with the lowest degree of freedom, quantum confinement of the electronic states occurs and introduces an entirely new class of physics to study the electron transport, reaction mechanism, and reaction rate. In the aspects of a resistive memory device, thin films that are prominently operated by the oxygen filament mechanism have several issues with reliability and performance. The most important one is the controllability of conductive filaments in the active dielectric medium, which is very crucial for device performance. We also need to consider the number of filaments formed (density) and the size of the filaments while talking about the endurance and reliability of the device. The next important issue is the scaling trend of metal oxide thin films, which is challenging when scaled down to the single-digit nanometer scale. Herein, quantum mechanics dominates and leads to uncertainty [24]. Whereas the low-dimensional materials fundamentally solve both issues and represent a logical pathway to the extreme scaling of semiconductor devices (by controlling the diameter of the nanowires), we have complete control over the conductive filament formation while retaining the device performance [25]. Already, many authors have reported the growth of *β*-Ga_2_O_3_ obtained via various methods, in that the VLS technique is considered a promising one, owing to the highly efficient control of the nanowire growth [22,26,27].

With such interesting practical applications on the horizon, we reported the comparison of CO gas-sensing applications of a pure and Au-decorated array of nanowires [22]. In this article, considering the significance and benefits of nanowires, we explored the fundamental resistive switching mechanism of a single *β*-Ga_2_O_3_ nanowire and discussed it in detail. The elemental compositions, intermediate absorption sites, and binding characteristics were confirmed by the energy-dispersive spectroscopy (EDS) spectrum from high-resolution transmission electron microscope (HR-TEM) analysis, cathodoluminescence (CL) from field-emission scanning electron microscope (FE-SEM) analysis, and X-ray photoelectron spectroscopy (XPS), respectively. The composition and binding energy analysis unveiled the crucial part of the oxide component in the fabricated metal oxide switching material. Finally, the switching mechanism of 1D nanowires is statistically discussed, and the physical mechanism is illustrated in detail.

## 2. Experimental Methods

Gallium oxide nanowires were obtained on Si (100) substrates using the vapor–liquid–solid (VLS) growing approach with Au nanoparticles as a catalyst. The substrates were cleaned according to the standard process of RCA cleaning. A thermal evaporator (Branchy Vacuum Technology Co., LTD., Taoyuan, Taiwan) system was used to form an 8 nm Au film on Si substrates. The annealing process was started when the substrate was heated to 900 °C at a 30 °C/min rate and held at 900 °C for 30 min to form Au droplets/nanoparticles (NPs), which would be our catalyst for the growth of nanostructures. In the aluminum boat placed in a horizontal quartz tube furnace (Lindberg/BlueM 1100 °C Tube Furnace, Asheville, NC, USA), a mixture of 0.05 g of gallium oxide powder (99.99% pure Analar grade, Alfa Aesar, Haverhill, MA, USA) and 0.075 g of carbon powder was added. The graphite powder was added into the reaction to reduce the decomposition temperature of the metal oxide, which usually has a very high evaporation temperature, and the detailed reaction mechanism on the alumina boat is given elsewhere [22]. Metal-catalyzed gold droplets on silicon substrates were kept in the horizontal tube furnace, as shown in Figure 1a.

From our optimization processes and further analysis, we found that keeping the distance between Au droplets/Si substrates and the source material (GaO + carbon powder mixture) equal to ~2 cm was the optimal position to result in 1D nanowires instead of obtaining a mixture of 1D and 2D triangular nanoleaf structures [16,24]. Argon gas was used as a carrier gas with a flow rate of 200 sccm. Once the carrier gas was passed, the entire system was then heated to 1025 °C at 30 °C/min. The nanowire samples were synthesized by maintaining the above temperature for 60 min, and as-grown nanostructures were naturally cooled to room temperature in order to leave the nanostructures to align to their equilibrium state. A schematic diagram of the nanowire growing process is shown in Figure 1b–d, and a detailed mechanism of the nanowire growth is given elsewhere [28].

## 3. Result and Discussion

### 3.1. Structural and Morphological Analysis

The preferred orientation for nanostructure growth and crystallinity was studied using grazing-incidence X-ray diffraction (GI-XRD) analysis using the X’Pert Pro MRD, PANanalytical instrument. Figure 2 shows the GI-XRD result of the synthesized sample. The diffraction peaks were matched with JCPDS card no 87-1901. The corresponding hkl values are indexed for *β*-Ga_2_O_3_. It is indicated that the prepared sample belongs to the monoclinic system of *β*-Ga_2_O_3_ with a preferred orientation along the ($\bar{2}02$) crystal plane, and the spiky diffraction peaks reveal the highly crystalline nature of the *β*-Ga_2_O_3_ [22,28]. The preferred orientation for the growth was confirmed by the high-intensity peak in the XRD spectrum, as well as in the real-space observation of the lattice plan orientation by HR-TEM.

FE-SEM (Zeiss Ultra Plus, Oxford Instruments) was employed to investigate the morphology. After the annealing treatment at 900 °C, it was observed that the gold droplets formed uniformly throughout the substrate with an average diameter of 77.74 ± 1.95 nm and, thus, acted as a catalyst. As mentioned in the experimental section, the sample kept ~2 cm away from the source compound resulted in the formation of a spindle wires (1D)-like structure. The FE-SEM image of 1D nanowires is shown in Figure 3a. Meanwhile, it was found that all the nanowires were uniform across the substrate. The average diameter of the nanowires was found by using Gaussian fitting of the calculated diameter of nanowires, which was equal to 144.96 ± 6.36 nm.

In order to study the single-nanowire structure, it was subjected to TEM (JEOL JEM-2010) analysis. The presence of Au catalyst nanoparticles on the top of a single nanowire was clearly visualized in the SEM image, as well as in the HR-TEM image shown in Figure 3b. The change in the size of Au nanoparticles on top of the nanowire after growth and the Au droplets formed on Si (100) substrate that guided the nanowire growth is attributed to the coalescence of Au droplets during nanowire growth, and it is a typical phenomenon observed in the VLS growth of metal oxide nanowires [29]. Figure 3c,d illustrate the high-resolution (HR) atomic lattice image and selected-area electron diffraction (SAED) pattern, respectively. The preferred nanowire growth plane ($\bar{2}02$) was observed in the HR-TEM image, and the same was witnessed in the SAED pattern, as well as the single-crystal nature of the grown nanowire samples. The obtained results of lattice constant values and corresponding Miller indices planes were assigned with the use of JCPDS data card No. 87-1901 as well [22,28].

### 3.2. Elemental Analysis

We employed EDS analysis from TEM to study the elemental composition of the prepared sample, depicted in Figure 3e–g. In order to explore the composition at a different region of the nanowire, we performed EDS over a selected area and elemental mapping. The TEM image of a single nanowire is shown in Figure 3b along with the markings R1 and R2, where the spectra shown in Figure 3e,f were taken, respectively. Figure 3g shows the Ga, O, and Au elemental mapping of the nanowire shown in Figure 3b. This confirmed the Au presence on top of the nanowire, which acted as a catalyst during the VLS process. From EDS analysis, it was observed that the gallium and oxygen atomic percentages for the 1D nanowire were 45.14% and 54.86% (taken at a local area of a nanowire, excluding Au), respectively. The atomic percentage ratio between Ga and O elements was around 2:3, merely coinciding with the standard value. However, it is worth noticing that the atomic percentage of oxygen was lower than the expected value, which is ascribed to the creation of oxygen vacancies during the high-temperature growth process.

### 3.3. Investigation of Bonding Features and RT-Cathodoluminescence Study 

The XPS delivered the chemical composition and the electronic binding state details of the prepared samples. The spectrum of as-grown 1D *β*-Ga_2_O_3_ nanowires on the silicon substrate is shown in Figure 4a–c. The results validated the characteristic signals from the prepared gallium oxide samples, which constitute Ga 3d, Ga 2p, O 1s, and the Auger electron peaks for both constituent elements. The Ga 2P_1/2_ and Ga 2P_3/2_ peaks are representative of Ga-O bonding. In order to visualize the different oxygen species, present in the nanowire lattices, we deconvoluted the O 1s peak and fitted it using a Gaussian function, and the same has been pictured in Figure 4c. This unveiled the three different oxygen species present in the sample, such as lattice oxygen (O_lat_/O^2−^), oxygen vacancies (O_vac_/O^–^), and surface-chemisorbed oxygen (O_ads_/O_2_^–^), which are the crucial factors considered for the performance of the device application. The labelling of each oxygen component has been adopted from the literature. The O 1s feature, which defines the state of oxygen in metal oxides, was divided into three components, (i) the oxygen ions around the metal ion with their full supplement of nearest-neighbor oxygen ions (fully oxidized: stoichiometric surrounding), labeled as O_lat_; (ii) oxygen ions that are in the oxygen-deficient regions within the lattice called O_vac_; here, the oxygen vacancy includes oxygen defects, interstitials, mis-orientation of lattice points, etc.; and (iii) loosely bound oxygen ions on the surface, mostly belonging to specific species, e.g., CO_3_ or adsorbed H_2_O, and O_2_, labeled as O_ads_ [29,30]. It is common in metal oxide materials that the percentage of oxygen vacancies could vary depending on the growth conditions, and it is always effectively fine-tuned for the proposed application. It is clear that the nanowires had a considerably high percentage of oxygen vacancies (43.5%), which was intentionally produced, and relatively low chemisorbed oxygen (3%). The rise in oxygen vacancies compared to our previous work was due to the change in the growth technique and parameters, which also favors the resistive switching application [30,31,32].

The room temperature-cathodoluminescence (RT-CL) spectrum acquired from *β*-Ga_2_O_3_ nanowires on the Si (100) substrate in the UV–visible range is shown in Figure 5. Herein, the spectrum from 250–700 nm is visualized. The JEOL JSM7001F SEM instrument with the CL system, which includes two spectrometers, was employed for this room temperature measurement. The spectrum shows a strong, broad peak in the UV-near-green emission region, three sharp peaks, and a weak peak in the green-red emission region. To see the contribution of all emission centers, we deconvoluted the RT-CL spectrum with Gaussian curves at wavelengths (energy) positioned at 332.98 nm (3.72 eV), 349.35 nm (3.55 eV), 389.18 nm (3.19 eV), 434.48 nm (2.85 eV), 488.62 nm (2.54 eV), 544.58 nm (2.28 eV), 585.09 nm (2.12 eV), 613.04 nm (2.02 eV), and 633.27 nm (1.96 eV). The first two peaks at 3.72 eV and 3.55 eV belong to UV emission, while the third at 3.19 eV corresponds to blue emission. The UV emission is ascribed to the recombination of self-trapped excitons. This transition feature is assigned to the bound excitonic luminescence from the transition of the ground state exciton level below the defect donor band to the valance band top edge. The deep blue emission is attributed to the recombination of electron–hole or donor–acceptor pairs (DAPs). Here, the donors are the oxygen vacancies, and the acceptors are the gallium vacancies. The high amount of oxygen vacancies introduced in the nanowires during the high-temperature growth process were validated from the CL results. The fourth (2.85 eV) and fifth (2.54 eV) peaks belong to cyan and near-green emission, respectively, and are assigned to the defect-to-defect transitions between the low-electron-density defect and the high-electron-density defect states. The low-electron-density defect and high-electron-density defect states refer to the defect energy levels located near the valance band and the conduction band energy levels, respectively. The latter one is a strong, sharp peak centered at 2.28 eV and corresponds to the green emission, which is also ascribed to the defect-to-defect transitions mainly from the defect clusters formed in the donor or acceptor energy levels. The last three peaks are ascribed to the red emission. These emissions bands are emitted from the low-energy defect states such as point defects and trap sites present in the deep near-band-edge levels. Several authors have described the origin of the above emission bands earlier in the literature [22,33,34]. In addition, we also observed several sharp and weak emission peaks around the green and red region, which is related to the recombination of electrons trapped by oxygen vacancies and holes captured by deep-level acceptor impurities such as nitrogen [33,34]. Additionally, trap energy levels in the single crystal *β*-Ga_2_O_3_ were reported, and it was found that there was a high density of trap and defect sites present above and below the Fermi energy level, which matches our CL measurement result [22,34]. It is worth mentioning that a considerable contribution from the base Si (100) substrate was added to the broad background signal in the CL spectrum [35].

### 3.4. ReRAM Device Characteristics

Figure 6a,b show the schematic diagram of the fabricated ReRAM device, along with the SEM image after the deposition of Au(50 nm)/Ti(10 nm) electrodes using a stainless-steel shadow mask. The device fabrication started with a uniform dispersion of nanowires on the SiO_2_/Si substrate exploiting the van der Waals force between the substrate and *β*-Ga_2_O_3_ nanowires, followed by the evaporation of metal electrodes with the help of a unilaterally placed shadow mask. Note that there was only one nanowire intersecting between the Au electrode pad pair, and a total of five devices were subjected to the ReRAM study, which displayed similar switching characteristics. The ReRAM device measurement was performed in a dark room at room temperature, and the current–voltage data were obtained by a Keysight B2912A precision source meter and measuring unit. The voltage sweep rate was kept at 0.04 V/s. Stable bipolar resistive switching without a forming curve was observed in the fabricated ReRAM device and is depicted in Figure 6c. In the 1D nanowire device, the continuous voltage loop of 0 V → +1.5 V → 0 V → −1.5 V → 0 V was used as the operating voltage. It is noted that the device was in the high-resistance state (HRS), also called the device OFF state, before the application of external bias voltage due to the high amount of oxygen vacancies and also the surface-chemisorbed atomic oxygen, which increased the insulating property. The compliance current was set to be 10 mA throughout our study. As the voltage increased from 0 V to 1.5 V, initially, the current started to increase linearly, and at ~1.4 V, the device was tuned to the low-resistance state (LRS), also called the SET process. The charge carriers easily overcame the Schottky barrier (~1.2 eV) created at the electrode/oxide contact at both terminals due to the densely populated intermediate trap energy levels, as observed in the CL spectrum. The SET voltage was 1.4 V (V_Set_). The device retained the LRS state during the sweep from 1.5 V to 0 V. When the voltage was swept from 0 to −1.5 V, the device switched back to the HRS at −1.15 V, also called the RESET process. The RESET voltage was −1.12V (V_Reset_). During the last voltage sweep from −1.5 V to 0 V, the current remained at the HRS state. From this complete ReRAM curve, we can conclude that the device exhibited stable bipolar resistive switching behavior.

The ON and OFF states can be read and reprogrammed to either state again in the subsequent SET and RESET processes, namely, both ON and OFF conditions are nonvolatile. This IV characteristic confirms that the fabricated devices could be used as a nonvolatile random-access memory device by implementing the bipolar “write-read-erase-read-rewrite” cycle. In order to study the longevity of the fabricated device, we recorded the continuous ON and OFF cycles for 200 cycles. The endurance cycle for 200 consecutive ON-OFF cycles is shown in Figure 6d, and it was found that the device deteriorated over many numbers of cycles; however, the LRS and HRS states were deliberately distinguishable even after 200 cycles. To validate the memory retention of the fabricated device, it was subjected to the retention test, and the result is shown in Figure 6e. The fabricated device exhibited excellent retention up to 10^4^ s, and the two distinct states then started to converge, showing the degradation of the nonvolatile nature of the memory cell. Even though the single nanowire had excellent chemical stability, consecutive set and reset processes eventually produced an inelastic change in the chemical state and caused a degradation in the resistance states. Lastly, the cumulative probability test to analyze the statistical distribution of HRS and LRS states in every cycle showed consistent ON and OFF states in each cycle, as depicted in Figure 6f. All the above performance study results exhibited excellent stability and reproducibility of the bipolar resistive switching (RS) behavior.

### 3.5. Physical Mechanism of Resistive Switching

The RS phenomenon is mainly interpreted in two ways: one is electrode-limited conduction through the nonfilament switching mechanism, and the other is switching material-limited conduction where the filament formation is the crucial part of the switching mechanism. As the electrodes used in our study were Au, which is not a reactive metal such as Cu or Ag, the metal breach into the *β*-Ga_2_O_3_ NWs was not possible and the electrode-dominated electrochemical metallization switching mechanism was excluded. The active material-dominated switching was categorized into valence change memory (VCM), thermochemical memory (TCM), and phase change memory (PCM). The TCM and PCM memory requires a significantly large electric field to produce switching through the Joule heating effect in the former and transformation between the crystalline and amorphous phase in the latter case, and both mostly resulted in the unipolar-type memory. Thus, the VCM was considered a suitable mechanism for our system, and other switching mechanisms were excluded. To verify the VCM switching, further analysis of our resistive switching curve was conducted as follows. The switching mechanism is more likely to be dominated by the filament formation mechanism based on our preliminary material characterization results of the switching material and also the detailed analysis of the bipolar switching curve. The fundamental resistive switching phenomenon of VCM metal oxide materials is explained by the formation and rupture of conductive filaments constructed by oxygen vacancies, defects, ion migration between electrodes through the grain boundaries, and so on [18,36,37,38]. Hsu et al. (2012) reported resistive switching through oxygen vacancy filament formation in a core–shell Au/Ga_2_O_3_ single nanowire, though a low density of oxygen vacancies in the Ga_2_O_3_ shell was perceived. In their study, no switching was observed in the pure single Ga_2_O_3_ nanowire, due to the inadequate oxygen vacancies [39]. In our investigation, an evident amount of oxygen vacancies in the *β*-Ga_2_O_3_ crystal lattices produced by our controlled growth process was confirmed. Hence, the oxygen vacancies dominated the formation and destruction of conductive filaments during the switching process.

Although the conductive filament formation and rupture accounted for the RS phenomenon, the fundamental electron transport during bias application needs to be clarified. There are several models to account for the conduction mechanism during the I–V sweep. In order to decide the correct model for our system, the obtained I–V curves for both SET and RESET processes were redrawn in a double log scale, as shown in Figure 7a,b. In the HRS state during SET and RESET processes, the current conduction can be differentiated through several regions due to the complexity. At the low-voltage region (0–0.18 V), the I–V curve was linear with a slope of ~1; at higher voltage (>0.19 V), the I–V curve was linear with a larger slope of ~2, and the slope promptly increased (>3.8) with the bias voltage. Then, the sample was switched from HRS to LRS. During the voltage sweep back to 0 V after the SET process, the slope was maintained at ~1.8, which suggests that all the traps remained filled, and the slope decreased to ∼1 with the decrease in the voltage as the carrier injection decreased again. However, it must be noted that many of the deep traps in the dielectric layer remained at the carrier trapped state, so the sample remained in the LRS until a sufficiently large bias voltage was applied. This progressing increase in the conduction (slope) is the dominant signature of the space-charge limited conduction (SCLC) mechanism.

The lower bias voltage indeed corresponds to the ohmic conduction through the free charge carriers that already exist in the material itself due to the weak injection of electrons from the metal electrode. At the transition voltage V_tr_, the voltage required to preside the transition from the ohmic to trap-filling process due to the strong injection of the electron, the injected electrons start to reorganize the free charge carriers in the material. At this point, the traps (oxygen vacancy, defects, etc.) activate and start to capture the free electrons, subsequently filling the traps. As the conductive filament formation is mainly dominated by the oxygen vacancies present in the Ga_2_O_3_ matrix, in the SCLC conduction, these oxygen vacancies act as the trap sites, which massively capture the injected electrons during bias application. This process is also recognized as trap-controlled SCLC (TC-SCLC). As the applied bias voltage crosses the voltage called the trap-filled limited voltage (V_TFL_), the I–V curve corresponds to the trap-free SCLC (TF-SCLC) [19,37]. During this period, all the traps are filled, and the excess electrons are free to conduct electricity between electrodes, so the conductive filament is formed to produce an LRS state. Meanwhile, the slope calculated in the LRS region is ~2 or smaller where more than one transport mechanism is involved along with SCLC conduction. Later, during the reverse bias application, the contact barrier at the electrode interface and grain boundary effect favorably causes the dissolution of the conductive filament to initiate; thus, the electrons trapped in the active sites become de-trapped and annihilate the electrode from where it was injected. Subsequently, the partial destruction of the conductive filament is achieved to produce the HRS state, which is reversible during consecutive voltage sweeps. The other possible conduction mechanisms in the LRS region and the dissolution of the conductive filament should be validated with further analysis that will be accomplished in our future study.

## 4. Conclusions

Through this study, we aimed to provide a detailed analysis of resistive switching of a single *β*-Ga_2_O_3_ nanowire solely from the growth of high-quality *β*-Ga_2_O_3_ nanowires to the fabrication and performance study of the ReRAM device. First, monoclinic *β*-Ga_2_O_3_ nanowires were grown using the VLS technique along with the Au catalyst. FE-SEM and TEM images validated the high density and homogeneity of the as-grown *β*-Ga_2_O_3_ nanowires. Besides, GI-XRD and SAED analysis confirmed that the prepared nanowires belonged to the monoclinic system with the preferred orientation of the ($\bar{2}02$) plane. The TEM EDS analysis confirmed that the Ga and O ratio was close to 2:3, which was in good agreement with the reported results. The XPS survey spectra illustrated the characteristic peaks of gallium and oxygen elements. Furthermore, the O1s core level scan explored the different oxygen species by deconvolution analysis. The cathodoluminescence study revealed the intermediate energy levels, which were consequences of the introduced oxygen vacancy defects in the high-temperature growth process of nanowires. The fabricated ReRAM device exhibited a stable I–V characteristic and ReRAM performance. The possible fundamental conduction mechanism and the physical switching mechanism were proposed.

## Figures and Tables

**Figure 1 nanomaterials-11-02013-f001:**
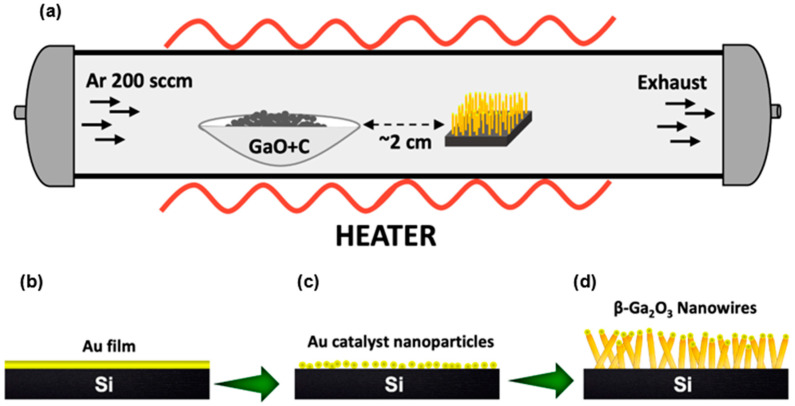
Schematic diagram for the growth of 1D β-Ga_2_O_3_ nanostructures on Si (100) substrate through the VLS method. (**a**) Entire setup during the growth process, (**b**) deposition of Au catalyst film on Si (100) substrate by PVD, (**c**) annealing of Au film to form Au eutectic droplets for nanostructure growth, and (**d**) 1D β-Ga_2_O_3_ nanowire growth at 1025 °C by VLS method guided by the Au nanoparticles.

**Figure 2 nanomaterials-11-02013-f002:**
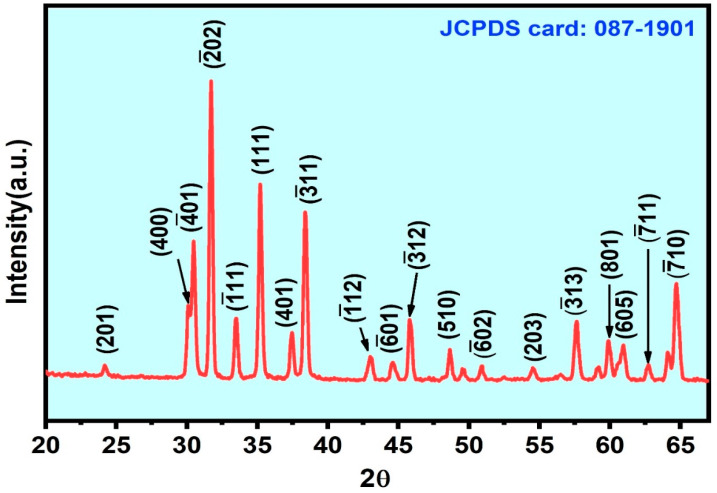
GI-XRD result of as-grown *β*-Ga_2_O_3_ nanowires on Si substrate.

**Figure 3 nanomaterials-11-02013-f003:**
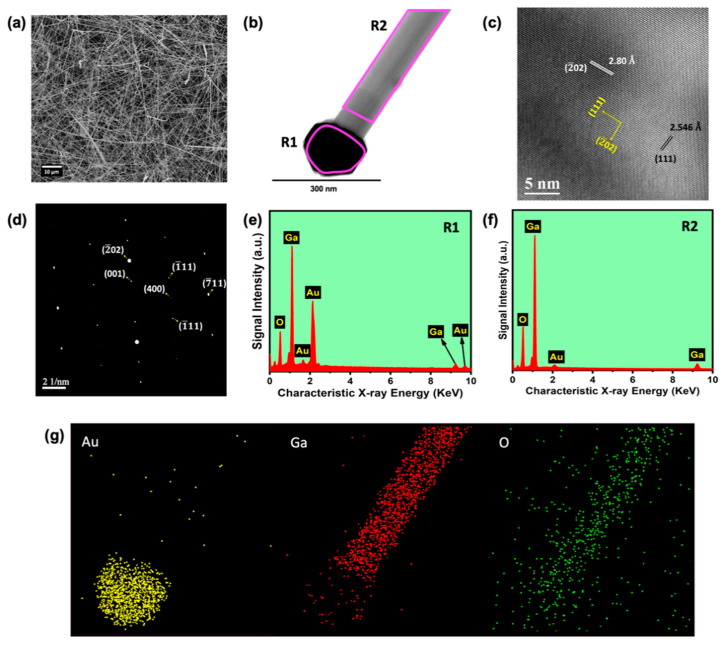
(**a**) SEM image of 1D *β*-Ga_2_O_3_ nanowires on Si (100) substrate, (**b**) TEM image of a single nanowire and Au catalyst nanoparticle on top of a single *β*-Ga_2_O_3_ nanowire, (**c**) HR-TEM image showing the crystal lattices, (**d**) SAED pattern of a single nanowire, (**e**) elemental mapping of individual Au, Ga, and O elements corresponding to the TEM image of a single *β*-Ga_2_O_3_ nanowire shown in (**b**), (**f**) EDS spectrum recorded at the marked regions R1 and R2 in the single nanowire shown (**g**).

**Figure 4 nanomaterials-11-02013-f004:**
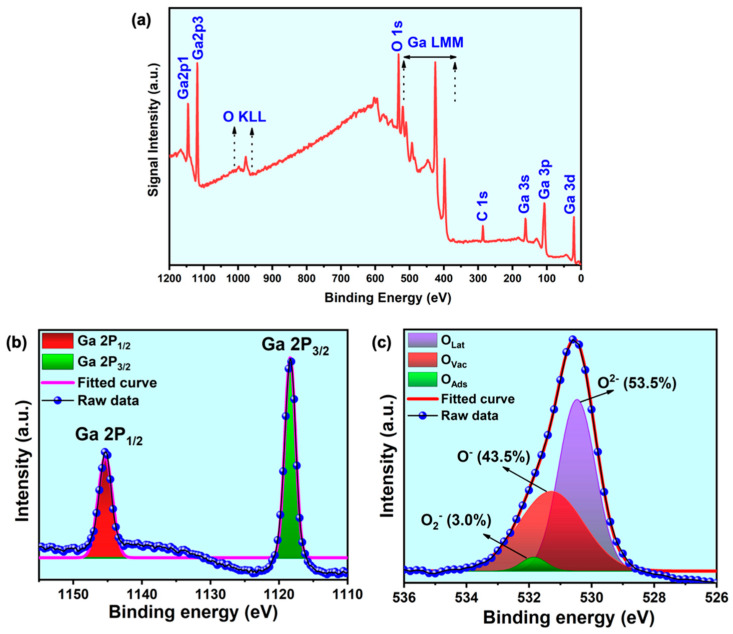
XPS spectrum of the as-grown *β*-Ga_2_O_3_ nanowires on Si (100) substrate. (**a**) Survey spectrum, (**b**) Ga2p core scan spectrum with Gaussian fitting, and (**c**) O1s core scan spectrum with Gaussian fitting showing the percentage of different oxygen species.

**Figure 5 nanomaterials-11-02013-f005:**
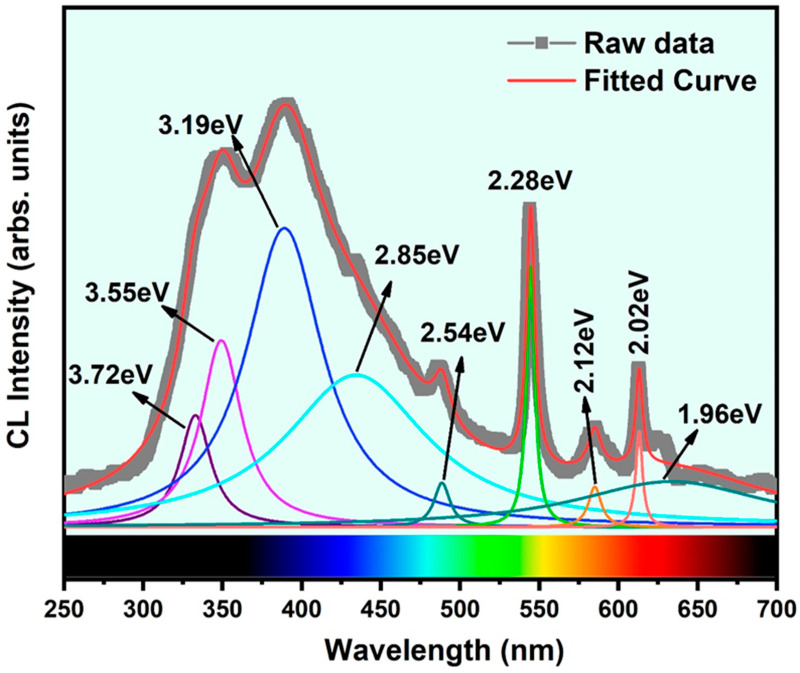
RT-cathodoluminescence spectrum recorded from the *β*-Ga_2_O_3_ nanowires grown on Si (100) substrate.

**Figure 6 nanomaterials-11-02013-f006:**
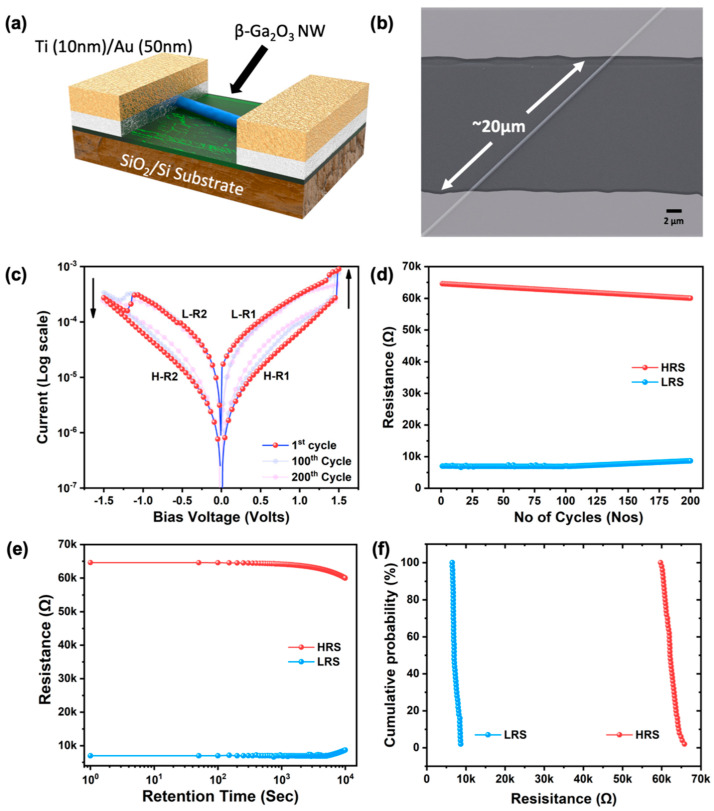
(**a**) Schematic diagram of the fabricated single β-Ga_2_O_3_ nanowire ReRAM device, (**b**) SEM image of the fabricated single *β*-Ga_2_O_3_ nanowire ReRAM device, (**c**) typical bipolar resistive switching curve of the single β-Ga_2_O_3_ nanowire, (**d**) endurance cycle results, (**e**) retention time analysis results, and (**f**) cumulative probability analysis of the fabricated device.

**Figure 7 nanomaterials-11-02013-f007:**
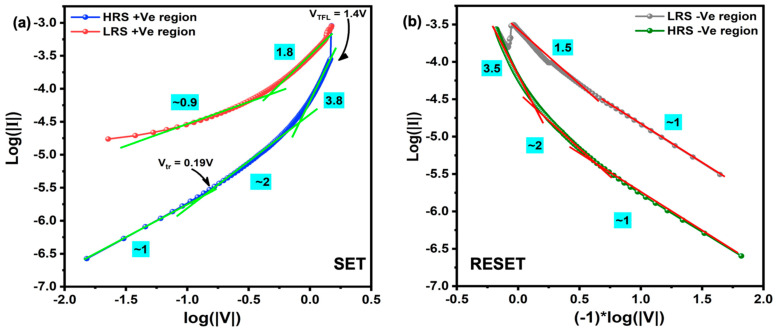
Double logarithmic plot of (**a**) SET and (**b**) RESET regions along with the slope for single-nanowire device.

## Data Availability

The data generated during and/or analyzed during the current study are available from the corresponding author on reasonable request.

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
