# Peer review of "High-Quality Single-Crystalline β-Ga2O3 Nanowires: Synthesis to Nonvolatile Memory Applications"

_nanomaterials, 2021, doi:10.3390/nano11082013_

Round 1
Reviewer 1 Report
The Authors reported the synthesis of high-density β-Ga2O3 NWs on Si substrates by VLS together with comprehensive characterization of the composition, structure, and properties of the obtained nanomaterials. Finally, the NWs were used in ReRAM devices and their performance is discussed in detail.
The topic of the manuscript is within the scope of the journal.
Abstract is informative enough. The introduction section describes the motivation and defines the research problem appropriately. The only suggestion would be to clearly state what is the element of novely with respect to the previous works published by the same authors on this topic.
The results are clearly described and discussed. The conclusions drawn are supported by the experimental results.
Therefore, I is my great pleasure to strongly recommend the manuscript for publication in Nanomaterials.
Please also correct some minor editorial and language issues.
Reviewer 2 Report
The manuscript addresses the synthesis, material analysis and resistive memory application of gallium oxide crystal wires. The subject is attractive in a field of metal-oxide non-volatile memory. The synthesis of the crystalline wires, characterization of crystal structure and chemical composition are well presented. The MIM device shows poor switching characteristics. A section on fabrication and characterization of ReRAM device is incomplete. The author’ discussion on the physical mechanism is not sufficiently supported by the experiment. There are multiple technical mistakes. I can not recommend to publish it in the present form.
Comments
- The authors decided to discuss only a mechanism involving growth of conductive filament without any explanation of the reason and substantial arguments. There are other physical mechanisms of resistive switching. Why did you choose this? Did you growth single-grain crystal wires without grain boundaries? A barrier at a grain boundary may explain the switching characteristics.
- Ga2O3 wires are a gas-sensitive and photo-sensitive material. What is the role of surface oxygen and the surface conductance? Does the I-Vs vary with photoexcitation? There are no considerations about it. Therefore, it makes the proposed mechanism doubtful.
- There are problems with MIM device stricture. Fig. 6 (a), the cartoon shows a wire under metal electrodes. However, the SEM image Fig.6(b) shows opposite – the wire on top of the metal stripes. It is inconsistent. Additionally, how did you control to have one Ga2O3 wire per MIM device? Stripe-shape electrodes can be crossed by many wires. Did the data in Fig.6 be taken on one-wire MIM? What is the wire diameter? How many devices did you measured?
- Fig.8 shows no difference in filament formation at positive or negative biases. Naturally, the filament can be formed either from bottom electrode, fig.8(b), or from top electrode, fig.8(c). The system is symmetric, and there is no clear reason to dissolve the filament at RESET voltage. The filament model is developed for thin film (smaller than 1 um) structures at high electric fields. The applicability of the filament model in your case is questionable, and must be discussed in details.
- The role of surface states at the wire surface and a barrier at electrode/wire contacts is unclear, and completely missing.
- You intended to show “the band alignment” in Fig.9. The figure is absent. Such figure will be helpful to understand the CL data in Fig. 5 and “Space-Charge Limited Conductance” as you stated on page 11.
- Why an average diameter of Ga2O3 wires (144.9 nm) is twice of that of Au dots (77.7 nm)? The Fig. 3(b) shows opposite.
- The reason to add carbon in gallium oxide powder is missing. It seems that amount of C atoms in the wires is large (XPS, Fig.4). It may affect the wire resistance. No discussion is present.
- Line 208, the statement “ … transitions between the low electron density defects to a high electron density defect state” is unclear. Please use common physical terms.
- Conditions and the instrumentation for the I-V measurement is missing. What is the voltage sweep rate?
Reviewer 3 Report
Summary
The authors demonstrate synthesis of single crystal beta-Ga2O3 nanowires of high crystalline quality (demonstrated by XRD and TEM). The nanowires were intentionally synthesized with a high density of oxygen vacancies, which is thoroughly demonstrated by XPS and CL analysis; this is important for their intended application as ReRAM dielectric medium. Finally, electrical characterization demonstrate that the nanowires can switch between high- and low-resistance states as required for ReRAM with good endurance and retention.
General comments
The key motivation for this study, as stated by the authors in the abstract, is: "low dimensional β-Ga2O3 nanowires are not explored in resistive memory applications, which hinders further developments.". However, I cannot find any explanation as to what specific problems (that exist for the more common thin-film) the nanowire geometry is supposed to address. This must be clarified.
Likewise, it is difficult to assess how the nanowire geometry is supposed to be beneficial to ReRAM devices. The authors list e.g. "high surface-to-volume ratio". But is this an advantage here? Might this not lead to more oxidation by air and thereby elimination of the desired oxygen vacancies? Furthermore, the benefit of "high-quality single-crystalline structure" is also unclear in this application: transport of oxygen vacancies along defects has been proposed to be beneficial for forming the conductive filaments in other materials (as I understand it). So the presence of defects/polycrystallinity is not necessarily detrimental. The specific benefits from nanowires, and how these benefits are evaluated from the experimental data, must be clarified.
In summary, the manuscript would benefit immensely from a clarified introduction which discusses the benefits of nanowires vs. 2D films, or single crystalline beta-Ga2O3 vs. polycrystalline or amorphous materials, for specifically ReRAM applications. Such an introduction would also help the reader understand the significance of the various characterization results that are presented, and make the novelty and significance much easier to assess.
Specific comments
Specific comments are listed by their line number.
46: What is the significance of doping to the ReRAM application? If doping is not used for ReRAM, then this can be left out. This goes back to my general comments that the manuscript would benefit from a clearer focus in the introduction to help the reader see the novelty and significance better.
121: I do not follow how the preferred (-202) orientation is derived from the XRD pattern, is it due to the high intensity of this peak compared to the others? This could be stated more clearly to aid the reader.
Figure 3b-d: the relative orientation of the overview and high-resolution image, and SAED pattern are unclear. This makes it difficult for the reader to assess the growth direction. As currently displayed, Figure 3b and c for instance show that the growth direction is (111), contrary to the description in the text. I think this is due to the images being rotated with respects to each other.
176: I do no think it is suitable to label a O 1s feature (which shows the presence of an oxygen atom) as an oxygen vacancy. It is an oxygen ion situated in a oxygen-vacancy-rich neighbourhood. It would be useful to provide a reference that discusses and motivates the assignment of these peaks in detail (which is not done in ref 16). It is difficult to find the origin of this assignment, maybe Journal of Applied Physics 48, 3524 (1977), but I would encourage the authors to clarify this description and referencing.
178: see above. The material cannot have >40% oxygen vacancies in the sense that 40% of the oxygen ions are missing (which is the most common understanding of the term "vacancy", as I see it). Please consider clarifying.
329: "Alongside, we have also derived the band alignments during the
switching processes, which has been shown in Figure 9a-b". No figure 9 is included as far as I can see.
Some additional minor language comments, by line:
64: "oxygen filament" should this be "oxygen vacancy filament"?
105: "operative" should this be "optimal"?
345: "TEM EDS analysis confirms the Ga and O ratio is 2:3" should be "is close to" instead?
Round 2
Reviewer 2 Report
The authors have made changes to the manuscript. However, some critical points have not been addressed adequately.
Previous comment - 7. Why an average diameter of Ga2O3 wires (144.9 nm) is twice of that of Au dots (77.7 nm)? The Fig. 3(b) shows opposite.
The authors gave explanation to the diameter difference based on the growth of ZnO wires. It is a special case. You need to provide with the evidence that similar growth mechanism was in your case of Ga2O3 wires. Add an image to Fig.3 to support it.
Previous comment -4. Fig.8 shows no difference in filament formation at positive or negative biases. Naturally, the filament can be formed either from bottom electrode, fig.8(b), or from top electrode, fig.8(c). The system is symmetric, and there is no clear reason to dissolve the filament at RESET voltage. The filament model is developed for thin film (smaller than 1 um) structures at high electric fields. The applicability of the filament model in your case is questionable, and must be discussed in details.
The authors gave extended explanation in the Reply copied from the literature. It seems that the author has wrong idea about the valence change memory (VCM) switching. It is “valence”, not “valance” (line 322).
There are many mistakes in the text and in the Reply. Most important, the authors gave no physical reason to observe the described mechanism – filament formation and dissolution – in the presented experimental system. The filament formation through vacancy migration can be observed under specific experimental conditions. In Ga2O3 materials the oxygen migration takes place at an electric field of greater than 50 000 V/cm, [see ref 2 in Reply]. In your case, the electric field is very low (560 V/cm =1.4 V/25um), and this condition is inconsistent with the migration threshold.
It is confusing that you are citing Ref. 2 in Reply to support your filament formation mechanism. The nanowires studied in Ref.2 have gold-core (ø40nm) / Ga2O3-shell structures, and results are not applicable to your wires. Actually, Ga2O3 wires without Au-core showed no switching or hysteresis in I-Vs below 50 V (600 000 V/cm) before burning. These results contradict your observations, and therefore, introduce a doubt on the proposed mechanism.
Moreover, you said on line 349 that “the slope decreased to ~1 “. I see a value of 0.65 at low voltage and 1.71 at high voltage in the graph. It is inconsistent with SCLC mechanism where the slope must be around “2” . Other data also did fit well to the parabolic dependence. Guo [ Appl. Phys. Lett. 106, 042105 (2015)] confirmed the Frankel-Poole conduction mechanism for Ga2O3 films. You may look at Electronics 2015, 4, 586-613; doi:10.3390/electronics4030586. It seems you did not take seriously to examine other mechanisms. Your conclusion about the mechanism is poorly supported and misleading.
It is also strange to have two different Log-scales for voltage in Fig.7. Did you applied Log(2.12) = 133 V?
Fig.9 was added to the manuscript. Though, I can hardly relate Fig.8 and Fig.9. You show that the oxygen vacancy moves to open defect conduction channel in Fig.8, but there is no additional conduction channel in Fig.9. What are black dots on top of AU electrodes? Why electrodes have 3 or 2 rows? Why defect levels are far above the E_F at the left electrode? Do the dotted arrows represent “electron hopping” over Schottky barriers? Does it represent the Mott hopping mechanism? I did not see much difference in “electron transport” in SET and RESET, it does not illustrate the mechanism.
In summary, the study of the switching mechanism is not developed adequately, and the authors may need additional measurements such as temperature dependence to clarify it. I suggest to reduce section 3.5 to outline the possible mechanisms, and leave the detailed analysis to future work considering my comments.
Round 3
Reviewer 2 Report
The authors have made important changes to the text. Nevertheless, mandatory corrections are needed.
Abstract. A statement on line 31-34 is incorrect. Please rewrite.
To Comment 1. The coalescence of Au NPs at nanowire growth conditions must be clearly stated with a reference in the text to avoid misunderstanding. Please add on line 164.
To Comment 2. Your point of view on a desirable switching mechanism is described in the Reply. However, it is in conflict with the main text.
(i) As you stated in the Reply : “we believe that a favorable switching mechanism would be the valence change (charge transfer) through the addition or removal of electrons (during current/electron flow over potential difference), also known as the redox reaction, which can also be described as electron hopping/transportation between localized oxygen vacancies along the conductive filament in the oxide layer.” According to your data in Fig 7, the SCLC conduction mechanism is not unique. Specifically, for the LRS transport, you have partial fitting at low voltage, and the slope is smaller than 2 (~1.8 for SET, and ~1.5 for RESET). Thus, other conduction mechanism is present. Because of insufficient data, you must write other possible explanation too.
(ii) You said on line 376 “during the reverse bias application, an apparent momentary Joule heating at the contact electrode interface initiates dissolution of conductive filament”. Surprisingly, you introduced a contact resistance switching mechanism, which differs from your proposed switching mechanism by “the formation and dissolution of conductive filament”. It is puzzling. I suggest, because of insufficient data to discuss, you must write clearly that other possible mechanisms – such as related to contact barriers and grain boundaries - may be involved, and further research is carried out.
(iii) You must discuss all relevant data such as opposite results for pure Ga2O3 NWs by Hsu (2012) to emphasis the role of vacancy defects in charge transport.
(iv) The English sentence is incorrect: “In our study, from the characterization results, it is witnessed that the existing oxygen vacancies in the β-Ga2O3 crystal lattices are attributed to the formation and destruction of conductive filament during the switching process.” (line 336-337) Please rewrite.
To Comment 4. The scale in Fig. 7(b) does not helpful for understanding the results. You calculated LOG(Abs(V)) , absolute value of Voltage. So, the x scale must be “(-1)*LOG(|V|)”.
